# Tissue Plasminogen Activator as a Possible Indicator of Breast Cancer Relapse: A Preliminary, Prospective Study

**DOI:** 10.3390/jcm11092398

**Published:** 2022-04-25

**Authors:** Katarzyna Wrzeszcz, Artur Słomka, Elżbieta Zarychta, Piotr Rhone, Barbara Ruszkowska-Ciastek

**Affiliations:** 1Department of Pathophysiology, Faculty of Pharmacy, Nicolaus Copernicus University, Collegium Medicum, 85-094 Bydgoszcz, Poland; katarzyna.wrzeszcz@abs.umk.pl (K.W.); artur.slomka@cm.umk.pl (A.S.); zarychta@abs.umk.pl (E.Z.); 2Clinical Ward of Breast Cancer and Reconstructive Surgery, Oncology Centre Prof. F. Łukaszczyk Memorial Hospital, 85-796 Bydgoszcz, Poland; rhonep@co.bydgoszcz.pl

**Keywords:** fibrinolytic profile, 5-year follow-up, invasive breast cancer

## Abstract

**Simple Summary:**

Breast cancer is the most common malignant neoplasm and the leading cause of cancer death in women worldwide. The heterogeneity of breast cancer is a significant challenge facing modern medicine, especially at the stage of diagnosis. In recent years, a substantial role of fibrinolytic biomarkers in the pathogenesis of breast cancer has been proved as the primary cause capable of initiating local recurrence and metastasis to other organs. Due to its proteolytic properties, fibrinolytic protein activity may lead to subendothelial layer degradation and promote neoplastic cell motility. Thus, we examined the fibrinolytic profile markers’ prognostic value in predicting the disease’s relapse. Our studies indicate that the t-PA antigen and PAI-1 activity are biomarkers with high prognostic potential. We showed that a lower baseline plasma concentration of t-PA antigen and higher PAI-1 activity might be strong predictors for distant metastases and independent prognostic markers in breast cancer patients.

**Abstract:**

(1) Background: The fundamental causes of breast cancer mortality are the cancer spread and hypercoagulability state. The study aimed to evaluate the prognostic efficacy of the fibrinolytic profile concerning 5-year follow-up. (2) Methods: We investigated the predictive potential of the plasma activity of urokinase plasminogen activator (u-PA) and plasminogen activator inhibitor type 1 (PAI-1) as well as antigen of tissue plasminogen activator (t-PA), u-PA, PAI-1, and PAI-1/t-PA and PAI-1/u-PA complexes in 41 breast cancer patients. The median follow-up was 66 months, with full evidence of the first event. (3) Results: A significantly lower level of PAI-1 antigen was noted in IBrC patients with lymph node involvement (N1) than in patients with free lymph node metastases (N0). According to ROC curve analysis, a t-PA antigen was the strongest predictor of disease relapse (the area under the curve, AUC = 0.799; *p* < 0.0006). Patients with PAI-1 activity < 3.04 U/mL had significantly better disease-free survival (DFS) compared to those with PAI-1 activity > 3.04 U/mL. Patients with both t-PA antigen lower than 1.41 ng/mL (cut-off according to median value) and lower than 1.37 ng/mL (cut-off according to ROC curve) had significantly shorter DFS (*p* = 0.0086; *p* = 0.0029). (4) Conclusions: The results suggest that a higher plasma t-PA antigen level or lower PAI-1 activity are linked to better outcomes in breast cancer patients.

## 1. Introduction

Invasive breast cancer (IBrC) is a relevant health issue associated with high global cancer morbidity and mortality among women [1]. Advances have been made in the molecular characterisation and development of novel targeted agents for the IBrC intrinsic subtypes; however, future perspectives for the advanced disease remain poor, with 5-year survival not extending beyond 28% in those cases [2].

It is well-established that the growth and dissemination of breast cancer are highly complex and multi-staged [3]. The data found in the literature emphasize that disturbances in haemostasis may play a fundamental role in the natural course of the disease [4,5]. Most researchers focus their attention on evaluating blood coagulation in patients with IBrC. These observations are consistent and have shown that patients with IBrC are at high risk of hypercoagulability state development, which significantly increases the likelihood of thromboembolic complications, additionally reinforced by the applied therapy [4,6,7,8,9]. The cancer-related prothrombotic threat is strongly linked with hypofibrinolysis expressed by diminished secretion of plasminogen activators and enhancement of its inhibitors [10].

Importantly, our studies [11,12] and other research teams [13,14] have shown that laboratory parameters determining blood coagulation could be diagnostic and prognostic factors in breast cancer. Such properties are attributed to, among others, tissue factor (TF) [11], von Willebrand factor (vWf) [12], and coagulation factors V (FV) [14] and FVIII [13].

Despite the advances in the knowledge of blood coagulation imbalances, the precise role of the fibrinolytic process in the pathophysiology, diagnostics, and prognosis of breast cancer remains obscure. Single clinical studies have mainly assessed the plasma levels of plasminogen activator inhibitor-1 (PAI-1), the primary fibrinolysis inhibitor [15,16,17,18]. It has been shown that PAI-1 may be highly clinically useful in the diagnosis and prognosis of breast cancer [17,18]. The role of the other elements involved in the blood clot liquefaction process is poorly understood, although Look et al. showed that tissue plasminogen activator (t-PA) and urokinase plasminogen activator (u-PA) [17], and Ruszkowska-Ciastek et al. [18] demonstrated that t-PA, might also be prognostic indicators in breast cancer.

Although the prognostic value of t-PA, u-PA or PAI-1 has been studied in various cancers, its particular value for prognosis evaluation in early-stage of IBrC has not been fully elucidated. There is a significant gap in the studies of the role of the fibrinolytic system in breast cancer, considering several laboratory parameters such as t-PA, u-PA, PAI-1, and especially the PAI-1/t-PA and PAI-1/u-PA complexes. Thus, the objective of the current study was to evaluate the prognostic efficacy of the fibrinolytic profile concerning 5-year follow-up.

## 2. Materials and Methods

### 2.1. Study Participants and Design

This study was carried out on a prospective series of 41 primary IBrC cases diagnosed between November 2015 and June 2017 by the Clinical Ward of Breast Cancer and Reconstructive Surgery, Oncology Centre in Bydgoszcz, Poland. All of the patients enrolled were women and received surgical and adjuvant treatment. All subjects were of Polish descent. All participants were interviewed by an oncologist with expertise in breast cancer, and blood samples were drawn for fibrinolysis parameters analysis between 2015 and 2017. The tumour, node, metastasis (TNM) classification, histological grade, oestrogen receptor (ER), progesterone receptor (PR), proliferative activity (Ki67), and human epidermal growth factor receptor 2 (HER2) status were provided by the Pathology Department of Oncology Center (Table 1). Tumour staging was defined according to the American Joint Committee on Cancer (AJCC; 7th ed.). Tumour size was evaluated according to the maximum tumour dimension.

### 2.2. Ethics

This study was performed based on the guidelines of the Declaration of Helsinki, and the Bioethics Committee of Collegium Medicum approved all procedures involving human subjects in Bydgoszcz, the Nicolaus Copernicus University in Toruń, Poland. Written informed consent was obtained from all participants before participating in the study.

### 2.3. Inclusion and Exclusion Factors

The essential inclusion criteria were primary, unilateral, invasive breast cancer in stage I maximum II. The exclusion criteria included (1) pre-operative findings of distant metastasis or another original tumour, (2) neoadjuvant chemotherapy or radiotherapy, (3) incomplete follow-up clinical data, (4) stage IIIA or higher, (5) grade 3, (6) carcinoma in situ, (7) tumour larger than 5 cm.

### 2.4. Study Outcomes and Follow-Up

Patients in the study population were followed from IBrC diagnosis to the date of breast cancer relapse or death or until 31 August 2021, whichever came first. Cumulative survival was presented by Kaplan–Meier graphs. During a median follow-up of 66.0 months (IQR 59–68 months), nine events occurred, including four (9.8%) distant metastases and five (12.2%) deaths. The recurrence rate was 22%.

### 2.5. Measurements

#### 2.5.1. Blood Sampling and Laboratory Tests

Venous blood from patients was drawn before treatment, processed, and immediately analysed for routine procedures. The blood was collected between 7:30 and 9:30 a.m. and after 12 h overnight fasting. Blood samples were collected into cooled tubes (Becton Dickinson Vacutainer^®^ System, Plymouth, UK) containing 0.13 mol/L trisodium citrate (final blood anticoagulant ratio 9:1). Samples were then mixed and centrifuged at 3000× *g* at +4 °C for 15 min, aliquoted and stored at −80 °C (as specified by the manufacturer) until assayed but no longer than 6 months. Storage conditions were carefully maintained, and aliquots were limited to one freeze–thaw cycle at batch analysis.

#### 2.5.2. Fibrinolytic Parameters

Commercially available enzyme-linked immunosorbent assay (ELISA) kits were used to analyse all fibrinolytic system parameters. Table 2 presents the specific features of those parameters concerning the name of the kit, manufacturer, assay range, detection limit, intra-assay coefficient of variation (CV, %) and inter-assay coefficient of variation (CV, %).

#### 2.5.3. Immunohistochemistry (IHC) Analysis

The estimation of oestrogen receptor (ER) and progesterone receptor (PR) status, expression of human epidermal growth factor receptor 2 (HER2) and Ki67-proliferation index utilized a standard procedure and was established by methods of immunohistochemistry (IHC). Hormone receptor status was defined as positive if at least 1% of tumour cells were noted with nuclear staining and negative if the nuclear staining was completely absent. HER2 status was classified as negative (score 0, 1+, and 2+ not amplified) or positive (when scored 3+ by IHC or HER2 amplified by fluorescence in situ hybridization (FISH)), according to the guideline for BC ASCO/CAP-2007. The Ki67 antigen was scored as a percentage of nuclei-stained cells of all the cancer cells by applying a monoclonal mouse antibody (Auto-stainer Link 48, Agilent Technologies, Santa Clara, CA, USA). For the Ki67 proliferation index, we used a 20% threshold as the limit to define high/low proliferative cases [12].

### 2.6. Statistical Analysis

All statistical analyses were conducted using Statistica v. 13.1 (StatSoft, Cracow, Poland). The Shapiro–Wilk test tested the normality of the data. Data are presented as medians and interquartile ranges (IQR) for non-normally distributed variables. Differences between groups were analysed using the Mann–Whitney U-test. The analysis was carried out to assess disease-free survival (DFS). DFS (in months) was calculated as the time frame between the date of surgery or the date of loco-regional/distant relapse (second IBrC, second primary cancer, and/or death without evidence of BC) to the date of the last contact. DFS curves were computed using the Kaplan–Meier method and compared using the log-rank test. The receiver operating characteristic (ROC) curves for all fibrinolytic parameters were plotted. The strength of associations was estimated by ROC analysis and expressed in terms of area under the ROC curve (AUC). AUC was calculated to assess the prognostic value. Optimal cut-off values were defined. Multivariate regression analysis was performed using Cox proportional hazards model. A multivariate Cox regression model included all variables with a significant effect in the univariate analysis to evaluate the independent impact of selected factors at the diagnosis on survival from breast cancer.

Additionally, the results were supported by further statistical analyses, including a univariate logistic regression analysis, which was used to predict the risk of developing breast cancer recurrence based on fibrinolytic factors. Multiple linear regression models with each fibrinolytic factor as an independent variable after adjusting for age, BMI, smoking, parity, menopausal status, tumour stage, tumour diameters, intrinsic type, histological type, and nodal involvement were also applied. The statistical significance level was *p*-value < 0.05.

## 3. Results

### 3.1. Recruitment and Participation

The basic characteristics of the participants are shown in Table 1. The average age of the participants was 56 years (IQR 51–60 years). The median BMI was 25.22 kg/m^2^ (IQR, 23.05–29.27 kg/m^2^). Tumour sizes ranged from 0.5 to 3.5 cm, with median size of 1.6 cm (IQR: 1.1–2.2 cm). Most of the tumours were <2 cm (24 cases, 59%). According to AJCC 7th edition for breast cancer, 16 patients (39%) were in stage I, and 25 (61%) were in stage II. The most common intrinsic/molecular subtype of breast cancer was luminal-A, confirmed in 26 subjects (63%), and the most common histological grade according to Elston-Ellis classification was G2, diagnosed in 31 cases (76%). The median follow-up duration for disease-free survival was 66 months (IQR, 59–68 months). Nine patients (22%) developed a relapse. Among them, three (7%) cases were diagnosed as luminal A subtype; in five (12%) subjects, luminal B HER2 (−) subtype was recognised, and only one (2%) patient had the luminal B-like HER2 (+) subtype of IBrC. None of the relapse patients had a triple-negative IBrC.

### 3.2. Clinical Presentation of Patients with Regard to Fibrinolytic Parameters

Table 3 presents variabilities in the antigen (concentration) and activity of fibrinolytic parameters concerning clinical and molecular characteristics. IBrC patients with lymph node involvement (N1) demonstrated a significantly lower level of PAI-1 antigen than patients with free lymph node metastases (N0) (*p* = 0.0412). The activity of PAI-1 was more than twofold higher in patients with luminal A IBrC than in other-molecular-types cases (*p* = 0.0410). However, there was a tendency toward a lower antigen of PAI-1 (*p* = 0.0565) in patients with grade 2 tumours compared to those with grade 1.

### 3.3. Fibrinolytic Parameters as Potential Markers of Breast Cancer Progression

The ROC curve was designed to estimate the prognostic value of the studied fibrinolytic parameters to predict the disease-free survival in breast cancer patients. The calculated areas under the curve (AUC) specified 95% confidence intervals. Based on the results of the assessment of the correctness of the investigated classifiers, the t-PA antigen was considered the strongest predictor of disease relapse (AUC = 0.799; *p* < 0.0006). Using the maximum value of the Youden index, t-PA antigen of 1.37 ng/mL with a specificity of 65.6% and a sensitivity of 88.9% was identified as the best cut-off value to distinguish patients with disease recurrence from those without disease relapse. The areas under the ROC curve for the PAI-1 activity, PAI-1 antigen, u-PA activity, u-PA antigen, PAI-1/u-PA complex, and PAI-1/t-PA complex were lower than the AUC for the t-PA antigen. Nevertheless, for all parameters assessed, the AUC was higher than the borderline of prognostic usefulness of the test ${\mathrm{AUC}}^{\mathrm{ROC}}$ > 0.5. Therefore, based on the Youden index, the cut-off point values were provided for all studied variables to discriminate between relapse-free patients and those with relapse (Table 4). Although the ${\mathrm{AUC}}^{\mathrm{ROC}}$ values for these mentioned parameters were above 0.5, the *p*-values were >0.05; thus, the strong prognostic power for the prediction of disease relapse was not reached (Figure 1).

### 3.4. Survival Analysis Regarding Fibrinolytic Parameters

All patients in this study received regular follow-up for 59 to 68 months (the median follow-up was 66 months). Overall, five patients (12.2%) died during the follow-up period due to systemic metastatic disease, and four (9.8%) cases developed relapse expressed by distant metastases. The calculated median and ROC cut-off point values of investigated fibrinolytic parameters were used to divide patients into two groups: the group with a baseline below the cut-off point and the group with a baseline above the cut-off value (Table 5). By applying the log-rank test, differences in the survival functions were observed only in the following variables: PAI-1 activity divided according to the ROC cut-off and t-PA antigen divided according to both cut-offs.

Patients with PAI-1 activity below 3.04 U/mL had significantly better DFS than those with PAI-1 activity above 3.04 U/mL (*p* = 0.0412), as shown in Figure 2B. Eleven patients (26.8%) demonstrated PAI-1 activity below 3.04 U/mL, whereas 30 patients (73.2%) had PAI-1 activity above 3.04 U/mL. Recurrence of the disease in the group of patients with PAI-1 activity below the cut-off point occurred in none of the 11 cases (0%), but in the group with PAI-1 activity above the cut-off point, 9 out of 30 (30%) patients had a recurrence of the disease.

According to Kaplan–Meier analysis for t-PA antigen with a cut-off point based on the median value, we observed that the survival rate was worse in patients with t-PA antigen below the cut-off point than in patients with t-PA antigen above the cut-off point. Twenty-one out of the 41 (51.2%) cases had t-PA antigen lower than 1.41 ng/mL; the recurrence rate for those subjects was 38.1% versus 5.0% for those with higher levels of t-PA antigen (Figure 3A). Similar results were also obtained for the t-PA antigen (Figure 3B) with the cut-off point based on the ROC cut-off point (*p* = 0.0029).

Based on Kaplan–Meier survival analysis, we postulate that higher PAI-1 activity or lower t-PA antigen concentration significantly correlates with an increased risk of breast cancer recurrence.

The remaining analysed parameters did not show any predictive value in our cohort (Figure 4A–D and Figure 5A–D). Nevertheless, the Kaplan–Meier analysis revealed an essential growing tendency towards a higher risk of disease relapse in patients with u-PA activity lower than 1.30 U/mL (*p* = 0.0969).

### 3.5. Potential Associations of the Disease-Free Survival with Fibrinolytic Parameters

A univariate logistic regression analysis was used to predict the risk of developing breast cancer recurrence as contributed by the fibrinolytic profile. Table 6 shows the odds ratio (OR) of recurrent breast cancer according to fibrinolytic parameters. It was found that only t-PA antigen significantly contributed to the disease-free survival. An increase of 1 ng/mL in t-PA antigen level was found to reduce the risk of breast cancer relapse by 0.06 (OR) (95% CI = 0.01 to 0.68; *p* = 0.0209). 

### 3.6. Analysis of Fibrinolytic Parameters as Prognostic Markers Using Univariate and Multivariate Cox Proportional Hazards Regression Models

In this study, we also used a univariate and multivariate Cox regression to analyse the prognostic factors of disease-free survival, which took into account the function of time. The multivariate Cox regression model was adjusted for prognostic factors including BMI, age at the time of diagnosis, smoking status, staging system, intrinsic type, histological type, nodal involvement and tumour diameter (Table 7). Thus, in a multivariate analysis, by including fibrinolytic parameters in a model with all the traditional predictive factors, we found no significant association of fibrinolytic parameter levels with the risk of breast cancer relapse (*p* > 0.05). However, the univariate Cox regression model showed that t-PA antigen was significantly associated with a prolonged DFS (HR = 0.10, 95% CI = 0.01–0.83, *p* = 0.0323). According to these results, patients with a t-PA antigen higher than 1.41 ng/mL appear to have a 90% decreased risk of breast cancer recurrence. 

### 3.7. Association of Fibrinolytic Parameters with Disease-Free Survival in Linear Regression Models

The associations of fibrinolytic parameter levels and disease-free survival by multiple linear regression analyses are shown in Table 8. Breast cancer recurrence was negatively associated only with t-PA antigen concentration using multivariate linear regression analyses after adjusting for age, BMI, parity, menopausal status and smoking status. In model 1 adjusted for age, the results showed that a lower t-PA antigen level was correlated with a higher risk of breast cancer relapse (standardized Beta = −0.4197; *p* = 0.0071). Similarly, in models 2 and 3, the results showed that a lower t-PA antigen level was associated with a higher risk of relapse occurrence after adjusting for age, BMI, parity, menopausal status and smoking status (standardized Beta = −0.3788; *p* = 0.0213 for model 2 and standardized Beta = −0.3815; *p* = 0.0227 for model 3, respectively). After adjusting for age, BMI, parity, menopausal status, smoking status, tumour stage, tumour diameters, intrinsic type, histological type, and nodal involvement (model 4), we found no significant trend for breast cancer relapse and t-PA antigen concentration (*p* > 0.05).

## 4. Discussion

Breast cancer is the most common malignant neoplasm and the leading cause of cancer death in women worldwide. Despite tremendous medical advances, the overall survival of patients with breast cancer has only slightly improved due to the vast heterogeneity of this neoplasm from the clinical, histopathological and molecular standpoints. Therefore, a new tactic and strategy should be adopted for a personalized approach to breast cancer patients. Such a new approach obliges contemporary medicine to search for effective predictive biomarkers that could be easily applied in everyday clinical practice. In recent years, the fibrinolytic parameters have been shown to play a significant role in the pathogenesis of several cancers as the main factors capable of initiating local recurrence and metastasis to other organs; thus, we hypothesize that parameters of the fibrinolytic profile, especially t-PA antigen, may have a significant role in breast cancer development and progression [1,10].

Biomarkers of the fibrinolytic system are essential factors in the haemostatic system involved in the dissolution of blood clots. In addition to their fundamental role in fibrin disintegration, the parameters of the fibrinolytic proteins also participate in many different pathophysiological processes. The leading cause of mortality from breast cancer is the development of metastasis to distant organs. Many factors play a crucial role in cancer dissemination, but vascular endothelial dysfunction is the fundamental cause of metastatic transmigration of cells. The endothelium produces most of the proteins involved in haemostasis processes, including tissue factor (TF), thrombomodulin, t-PA, u-PA or PAI-1. The proteolytic properties of the fibrinolytic proteins affect the fibrin dissolution and lead to the disintegration and degradation of the main components of the vascular endothelial basement membrane, which in turn promotes the neoplastic cell migration. The latest reports indicate that abnormalities of the coagulation and fibrinolysis processes are linked with neoplastic disease and indirectly lead to tumour progression and development. In the light of new data, t-PA is required for promotion of tumour growth, angiogenesis and its invasive phenotype [18,19,20,21,22,23]. 

We decided to estimate the prognostic value of plasma selected fibrinolytic factors in patients with early-stage breast cancer. This study indicated that t-PA antigen may serve as an independent biomarker for the risk of disease recurrence or dissemination in early-stage breast cancer. We established that plasma levels of t-PA antigen lower than 1.41 ng/mL were associated with a poorer prognosis and shorter DFS. Several other statistical analyses confirmed this observation. According to the ROC curve and the AUC value, we showed that t-PA antigen is the strongest predictor of disease relapse among all investigated fibrinolytic markers (AUC = 0.799; *p* < 0.0006). Based on the Youden index, we pointed out that the cut-off point of 1.37 ng/mL of t-PA antigen may serve as a value that discriminates between disease recurrence patients and those without disease relapse, with a specificity of 65.6% and sensitivity of 88.9%. Using the multivariate Cox regression model and logistic regression analysis, it was ultimately found that a low baseline concentration of t-PA correlated with poorer disease-free survival. The lower level of t-PA antigen was associated with a more enhanced risk of breast cancer progression. It can be assumed that patients with a baseline higher t-PA concentration (>1.41 ng/mL) are 90% less prone to develop breast cancer recurrence than patients with a lower baseline t-PA concentration. Thus, we speculate that the downregulation of t-PA concentration is associated with worse outcomes in breast cancer patients. Our current research provides strong power and evidence because it has been confirmed by using advanced statistical methods including Kaplan–Meier survival analysis or Cox regression, logistic regression, and multivariate linear regression. Similar results have been obtained by Corte et al., who proved the negative relationship between the expression of t-PA in breast carcinoma tissue and prognosis. According to them, lower t-PA levels in breast cancer patients are associated with a shortened relapse-free and overall survival. The authors take advantage of two facts: t-PA is an oestrogen-inducible enzyme, and the presence of oestrogen receptors (ER) is generally correlated with a good outcome. They hypothesize that the concentration of t-PA may directly reflect ER’s individual and biological activity [24].

Despite this fact, we would like to refer to the current results of our previous study [18]. Surprisingly, we obtained completely opposite results in an earlier investigation since a higher concentration of t-PA was linked with a more aggressive tumour character and a higher possibility of breast cancer dissemination. Additionally, a marked tendency toward a higher concentration of t-PA in T2, stage IIA+IIB, and lymph node-involved tumours was noted. Divergent results may be caused by different sizes of the study population, cut-off points or differences in the reagent kits used for measurement that were manufactured by other companies. Therefore, establishing adequate methodology and cut-off points for fibrinolytic factors are relevant to reaching a consensus. Costanzo et al. also revealed results inconsistent with those of our current study, since the authors reported that an elevated concentration of plasma t-PA antigen is a positive and independent biomarker of breast cancer risk. The authors suggested that a higher concentration of t-PA may lead to the development of metastasis due to proteolytic degradation of the extracellular matrix. Under these conditions, t-PA acts as a molecular mediator of angiogenesis. The plasminogen activator/plasmin system, due to enzymatic cascade, is involved in the conversion of plasminogen to plasmin. Plasmin then degrades the main components of the vascular endothelial basement membrane and activates matrix metalloproteinases that degrade the extracellular matrix, allowing cancer cells to migrate, invade and spread to distant sites. Moreover, Costanzo et al. imply that t-PA is a potent inductor of cell proliferation, including breast cancer cells [25]. Similar results have been presented in the study performed by Chernicky et al. The authors proved that low levels of t-PA are correlated with a good prognosis and more extended disease-free survival periods. This study explains that breast cancer cells express the insulin-like growth factor receptor and respond to insulin-like growth factor (IGF) in the environment. Based on this fact, Chernicky et al. revealed that IGF plays a role in upregulating t-PA levels that may contribute to breast cancer’s aggressive behaviour. Moreover, the authors showed that high u-PA levels indicated a poor prognosis in a breast cancer patient, which was not confirmed in our study [26]. However, the expression or concentration of fibrinolytic parameters may differ between various tissues and cancer types, and those discrepancies warrant further studies in this regard. Despite the continual advancement of knowledge, a consensus on this point has still not been reached.

Several studies indicate that u-PA concentration is a more accurate diagnostic and prognostic biomarker than t-PA [17,27]. Additionally, there is evidence that cancer invasion and metastasis mainly depend on u-PA and its receptor activity [28]. However, such observation is inconsistent with our results. Since applying numerous advanced statistical tests apart from Kaplan–Meier analysis, the results regarding u-PA antigen and activity did not differ significantly. The Kaplan-Meier analysis revealed an essential growing tendency towards a higher risk of disease relapse in patients with lower activity of u-PA than 1.3 U/mL (*p* = 0.0969). This observation identified a similar trend of u-PA activity as the t-PA antigen. Most likely, the lack of statistical difference may be due to the small size of the study group. The study also indicated that patients with early-stage breast cancer more frequently demonstrate lower u-PA activity (33 cases (80%) with lower vs. 8 (20%) with higher u-PA activity than 1.30 U/mL).

The blood was collected in standard conditions to limit the impact of circadian lability of fibrinolytic parameters. Nevertheless, shift work, work at night, or chronic jet lag may impact the main fibrinolytic agents’ circadian secretion. Interestingly, circadian haemostatic factor expression may promote the hypercoagulability phenotype by elevating PAI-1 and reducing t-PA [29]. Surprisingly, in our investigation, the concentration of PAI-1 antigen was significantly lower in breast cancer patients with lymph node involvement than in tumour-free lymph node patients. Moreover, according to the molecular classification, the subgroup of luminal-A breast cancer patients displayed PAI-1 activity more than twice as high as that in other-molecular-types cases. However, it is worth noting that patients with PAI-1 activity below 3.04 U/mL had a significantly better DFS. Despite this inconsistency, our current and previous studies suggest that PAI-1 may predict disease relapse; thus, PAI-1 expression presents prognostic and predictive value in breast cancer patients [18]. Interestingly, no fibrinolytic markers other than PAI-1 activity and antigen showed potential association with tumour diameter, TNM classification, histological grade, oestrogen receptor expression, progesterone receptor expression and Ki67 activity. This may suggest that none of these other factors significantly influenced the fibrinolytic parameter profile. Despite our findings, many authors provided strong evidence that u-PA, PAI-1 and u-PA/PAI-1 are vital factors for tumour invasion and metastases in breast cancer [27,28]. Lampelj et al. established that high levels of both proteolytic enzymes u-PA and PAI-1 are related to poor prognosis in patients with breast cancer [30]. Further, Harbeck et al. confirmed these observations [31]. The authors explain that u-PA plays a significant role in proteolytic degradation of the extracellular matrix, enabling the development of metastases. Furthermore, PAI-1 has a crucial and paradoxical role in cancer-promoting angiogenesis and tumour cell survival. Despite that, PAI-1 is an inhibitor of u-PA in the plasminogen activator system, and theoretically, PAI-1 should be anticipated to have an anti-tumorigenic function; surprisingly, PAI-1 mediates in adhesion, migration, invasion, proliferation and apoptosis of normal and malignant cells through its anti-protease and vitronectin-binding functions and cancer cell apoptosis inhibition as well [30,32]. Based on these contrary results, further research is needed. These discrepancies seem surprising, but it is worth noting that there are considerable differences in the collection of the biological samples used to measure t-PA. Some of the available studies assessing the predictive value of the fibrinolytic parameters in breast cancer focus on the measurement of these biomarkers’ expression only in the neoplastic tumour environment. The material for research in most available sources was mainly neoplastic tissue (tumour) [24], while in this study, the concentrations of fibrinolytic parameters were assessed only in plasma. In addition, the use of adjuvant therapy could have impacted the fibrinolytic parameters [10].

### Limitations of the Study

There are some limitations to our study. First, we included a small number of patients, which could constitute a bias; data should be confirmed in a more extensive and prospective series of patients. Although we collected and adjusted data for probable confounders in our study, there could be unmeasured and undefined factors with possible residual effects. We evaluated patients in single-centre designed research. Further prospective studies with more significant patient numbers and longer follow-ups are required to investigate the long-term outcome. Sufficient evidence has not yet been accumulated to determine cut-off values of the t-PA, u-PA, PAI-1 or PAI-1/t-PA and PAI-1/uPA complexes for early-stage IBrC; thus, further evaluation should be performed.

## 5. Conclusions

Despite the small cohort size, we suggest that a higher plasma t-PA was linked with a good prognosis and that t-PA antigen demonstrates substantial predictive value since this observation was confirmed by the application of several advanced statistical tests. Additionally, lower PAI-1 activity was associated with better outcomes in breast cancer patients. We hypothesize that in those cases with a higher t-PA level or lower PAI-1 activity, it is a possible enhancement of the fibrinolysis process, which is associated with better prognosis. Nevertheless, that hypothesis should be confirmed in a larger, adequately powered study.

## Figures and Tables

**Figure 1 jcm-11-02398-f001:**
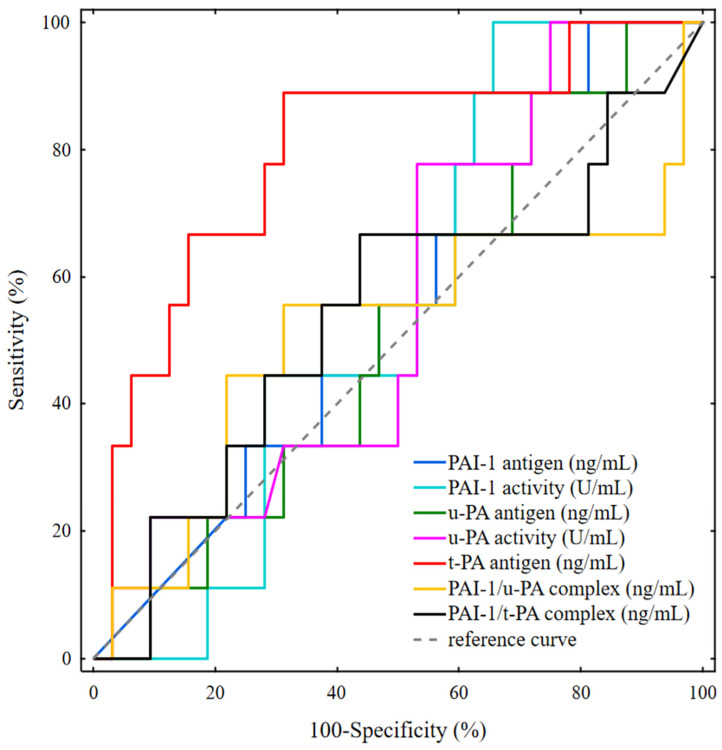
ROC curve analysis of the investigated fibrinolytic biomarkers.

**Figure 2 jcm-11-02398-f002:**
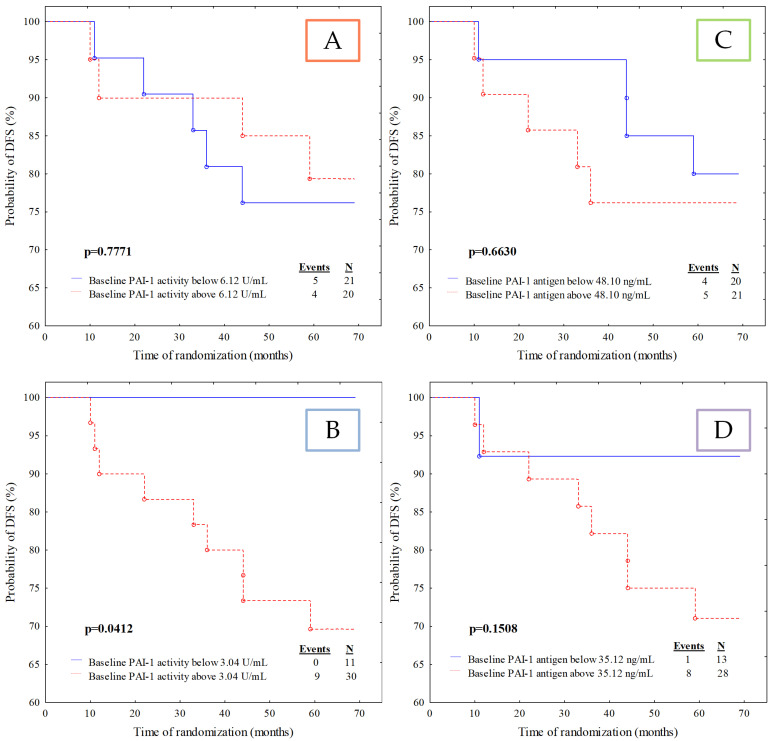
Kaplan–Meier curves for the disease-free survival (DFS) analysis of the studied population regarding (**A**) PAI-1 activity divided according to median value cut-off; (**B**) PAI-1 activity divided according to ROC cut-off; (**C**) PAI-1 antigen divided according to median value cut-off; (**D**) PAI-1 antigen divided according to ROC cut-off.

**Figure 3 jcm-11-02398-f003:**
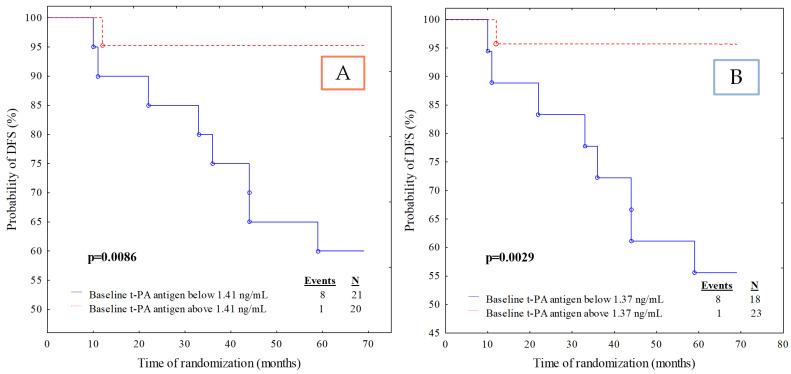
Kaplan–Meier curves for the disease-free survival (DFS) analysis of the studied population regarding (**A**) t-PA antigen divided according to median value cut-off; (**B**) t-PA antigen divided according to ROC cut-off.

**Figure 4 jcm-11-02398-f004:**
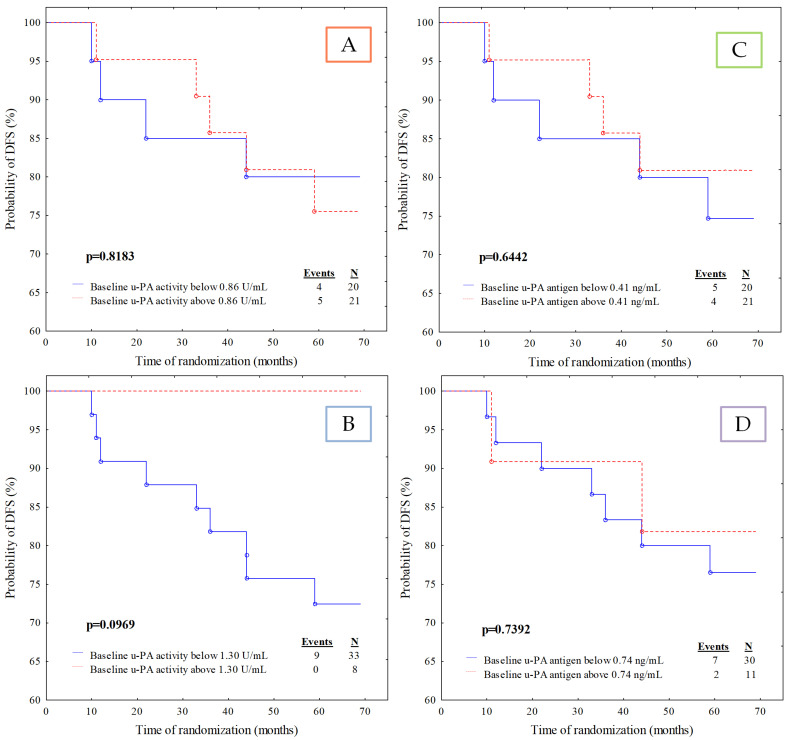
Kaplan–Meier curves for the disease-free survival (DFS) analysis of the studied population regarding (**A**) u-PA activity divided according to median value cut-off; (**B**) u-PA activity divided according to ROC cut-off; (**C**) u-PA antigen divided according to median value cut-off; (**D**) u-PA antigen divided according to ROC cut-off.

**Figure 5 jcm-11-02398-f005:**
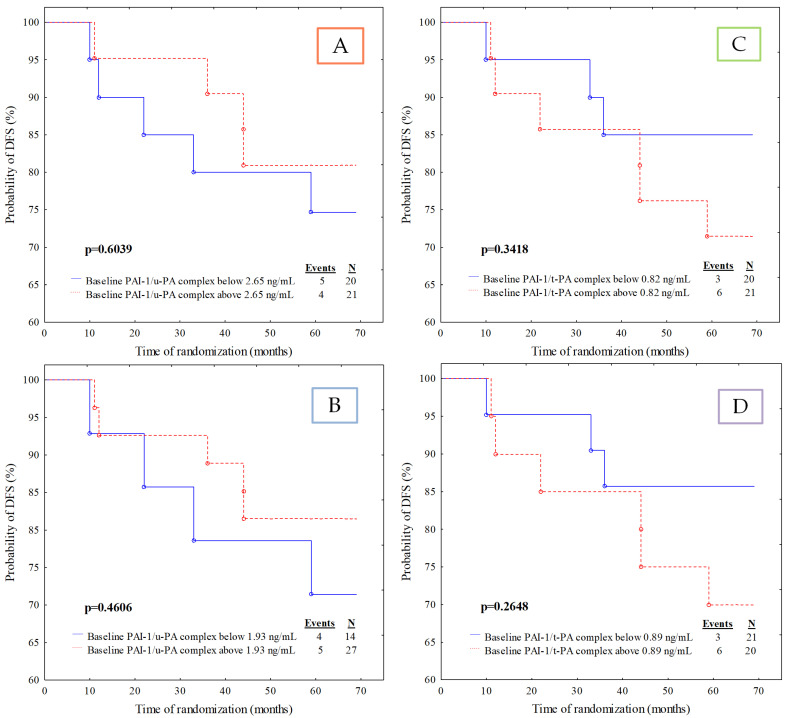
Kaplan–Meier curves for the disease-free survival (DFS) analysis of the studied population regarding (**A**) PAI-1/u-PA complex divided according to median value cut-off; (**B**) PAI-1/u-PA complex divided according to ROC cut-off; (**C**) PAI-1/t-PA complex divided according to median value cut-off; (**D**) PAI-1/t-PA complex divided according to ROC cut-off.

**Table 1 jcm-11-02398-t001:** Patient characteristics.

Demographic and Clinical Data	Total (n = 41; 100% of European Ancestry)
Age (according to median)	
<56	18 (44%)
≥56	23 (56%)
Menopausal status	
pre-menopause	12 (29%)
post-menopause	29 (71%)
T status (7th ed.)	
T1	24 (59%)
T2	17 (41%)
N status (7th ed.)	
N0	28 (68%)
N1	13 (32%)
Stage (7th ed.)	
IA	16 (39%)
IIA+IIB	25 (61%)
Molecular components	
ER (+)	37 (90%)
ER (−)	4 (10%)
PR (+)	33 (80%)
PR (−)	8 (20%)
HER2 (+)	5 (12%)
HER2 (−)	36 (88%)
Proliferation marker expression	
Ki67 < 20%	29 (71%)
Ki67 ≥ 20%	12 (29%)
Histological grade	
1	10 (24%)
2	31 (76%)
Histological type	
Ductal	35 (85%)
Lobular	6 (15%)
Tumour localisation	
Left breast	19 (46%)
Right breast	22 (54%)

N0: lack of lymph node metastases; N1: spread to axillary lymph nodes; T1: tumour diameter ≤ 2 cm; T2: tumour diameter > 2 cm to ≤5 cm; ER: oestrogen receptor; PR: progesterone receptor; HER2: human epidermal growth factor receptor 2; Ki67: proliferation marker; G1: low grade; G2: moderate grade.

**Table 2 jcm-11-02398-t002:** The basic quality of enzyme-linked immunosorbent assay (ELISA) kits used in the present study.

Kit	Manufacturer	Assay Range	Detection Limit	The Intra-Assay Coefficient of Variation (CV, %)	The Inter-Assay Coefficient of Variation (CV, %)
Human PAI-1 Total Antigen ELISA Kit	Molecular Innovations, Inc., Novi, MI, USA	0.25–100 ng/mL	0.122 ng/mL	3.13–7.89	3.83–8.10
Human PAI-1 Activity ELISA Kit		0.125–100 U/mL	0.113 U/mL	4.74–9.18	7.85–9.52
Human u-PA Total Antigen ELISA Kit		0.1–50 ng/mL	0.019 ng/mL	2.77–7.16	<15
Human u-PA Activity ELISA Kit		0.1–50 ng/mL	0.013 U/mL	1.60–6.27	5.62–7.37
Human t-PA Total Antigen ELISA Kit		0.2–25 ng/mL	0.0108 ng/mL	<10	<15
Human PAI-1/u-PA 1 Complex Antigen ELISA Kit		0.1–100 ng/mL	0.070 ng/mL	<10	<15
Human PAI-1/t-PA Complex Antigen ELISA Kit		0.5–100 ng/mL	0.03 ng/mL	<10	<15

PAI-1: plasminogen activator inhibitor-1; t-PA: tissue plasminogen activator; u-PA: urokinase plasminogen activator.

**Table 3 jcm-11-02398-t003:** The fibrinolytic profile analysis with respect to clinicopathological features.

Variables of Interest	PAI-1 Activity [U/mL] *p*-Value	PAI-1 Antigen [ng/mL] *p*-Value	t-PA Antigen [ng/mL] *p*-Value	u-PA Activity [U/mL] *p*-Value	u-PA Antigen [ng/mL] *p*-Value	PAI-1/u-PA Complex [ng/mL] *p*-Value	PAI-1/t-PA Complex [ng/mL] *p*-Value
N0 N1	6.32 1.65/13.12 6.05 3.31/9.50 *p* = 0.9442	51.24 35.81/100.00 29.33 23.76/48.36 ***p* = 0.0412**	1.51 1.20/1.75 1.33 1.12/1.57 *p* = 0.2324	0.79 0.60/1.23 0.98 0.67/1.20 *p* = 0.7474	0.41 0.24/0.73 0.47 0.27/0.74 *p* = 0.9442	2.58 1.51/3.88 2.98 1.66/5.65 *p* = 0.2565	0.73 0.30/1.24 0.99 0.46/1.69 *p* = 0.6042
T1 T2	5.51 1.67/13.07 7.58 3.39/10.83 *p* = 0.5694	48.97 28.44/91.64 42.18 29.56/91.67 *p* = 0.9052	1.55 1.25/1.73 1.32 0.98/1.52 *p* = 0.1216	0.92 0.72/1.23 0.78 0.55/1.03 *p* = 0.3210	0.44 0.23/0.74 0.40 0.28/0.72 *p* = 0.9262	2.77 1.54/4.51 2.60 1.62/5.23 *p* = 0.8015	0.90 0.32/1.71 0.67 0.33/1.18 *p* = 0.5168
Stage I Stage II	6.32 1.48/19.41 3.05 3.31/9.50 *p* = 0.9680	58.03 40.84/100.00 39.03 25.37/68.37 *p* = 0.0931	1.55 1.34/1.75 1.33 0.98/1.57 *p* = 0.1606	0.84 0.72/1.45 0.91 0.55/1.06 *p* = 0.2779	0.38 0.23/0.79 0.47 0.27/0.72 *p* = 0.9893	2.58 1.54/3.76 2.96 1.62/5.65 *p* = 0.3565	0.79 0.25/1.47 0.89 0.46/1.27 *p* = 0.9255
Grade 1 Grade 2	3.19 1.33/14.52 6.92 3.31/11.74 *p* = 0.2189	85.80 45.18/100.00 41.94 25.37/68.37 *p* = 0.0565	1.48 1.22/1.73 1.41 1.10/1.68 *p* = 0.7488	0.92 0.73/1.57 0.86 0.55/1.06 *p* = 0.0904	0.47 0.32/0.84 0.40 0.25/0.74 *p* = 0.5744	1.86 0.89/4.26 2.94 1.62/5.65 *p* = 0.1767	0.90 0.23/1.73 0.82 0.33/1.27 *p* = 0.8198
ER (+) ER (−)	6.12 3.04/11.74 4.56 1.16/10.57 *p* = 0.4419	48.10 27.54/83.27 66.19 34.52/96.68 *p* = 0.5219	1.41 1.12/1.68 1.77 1.04/2.46 *p* = 0.1156	0.91 0.64/1.20 0.75 0.59/1.21 *p* = 0.7088	0.39 0.25/0.72 0.73 0.57/0.93 *p* = 0.5531	2.56 1.59/4.59 3.88 3.05/5.41 *p* = 0.2105	0.89 0.27/1.27 0.71 0.62/1.99 *p* = 0.7584
PR (+) PR (−)	6.12 3.31/12.68 5.16 1.62/8.40 *p* = 0.3157	48.10 27.54/91.67 46.06 34.52/73.28 *p* = 0.9604	1.41 1.12/1.68 1.39 1.04/2.19 *p* = 0.3189	0.93 0.67/1.26 0.73 0.53/0.87 *p* = 0.2429	0.37 0.25/0.74 0.57 0.45/0.80 *p* = 0.8051	2.50 1.59/4.59 3.37 2.58/5.41 *p* = 0.3656	0.89 0.27/1.28 0.71 0.45/1.17 *p* = 1.0000
Ki67 < 20% Ki67 ≥ 20%	6.12 1.79/11.74 6.66 2.67/11.39 *p* = 0.9886	45.18 29.33/68.37 60.77 26.88/96.68 *p* = 0.5941	1.52 1.29/1.68 1.24 1.01/1.75 *p* = 0.6432	0.96 0.55/1.26 0.76 0.70/1.07 *p* = 0.6364	0.39 0.26/0.67 0.63 0.30/0.81 *p* = 0.3980	2.89 1.62/4.59 2.58 1.41/4.75 *p* = 0.8974	0.89 0.38/1.27 0.71 1.41/4.75 *p* = 0.7526
Luminal A Other types	8.26 3.31/13.46 3.53 1.58/7.27 ***p* = 0.0410**	45.02 25.37/71.60 49.84 38.84/100.00 *p* = 0.2948	1.45 1.29/1.68 1.18 0.90/1.78 *p* = 0.8989	0.81 0.56/1.06 0.93 0.73/1.30 *p* = 0.6357	0.32 0.22/0.67 0.59 0.40/0.88 *p* = 0.0530	2.60 1.59/4.59 3.24 1.64/6.56 *p* = 0.5072	0.77 0.27/1.21 0.89 0.56/1.73 *p* = 0.3863

PAI-1: plasminogen activator inhibitor-1; t-PA: tissue plasminogen activator; u-PA: urokinase plasminogen activator; N0: lack of lymph node metastases; N1: spread to axillary lymph nodes; T1: tumour diameter ≤ 2 cm; T2: tumour diameter > 2 cm to ≤5 cm; ER: oestrogen receptor; PR: progesterone receptor; HER2: human epidermal growth factor receptor 2; Ki67: proliferation marker; G1: low grade; G2: moderate grade; Data are expressed as median (Me) and the inter-quartile range (IQR) [lower quartile (Q1)/upper quartile (Q3)]; bold *p*-values denote significant differences.

**Table 4 jcm-11-02398-t004:** Results of predictive accuracy for individual fibrinolytic parameters.

ROC Data	Stimulant	Stimulant	Destimulant	Destimulant	Destimulant	Destimulant	Stimulant
	PAI-1 Activity	PAI-1 Antigen	t-PA Antigen	u-PA Antigen	u-PA Activity	PAI-1/u-PA Complex	PAI-1/t-PA Complex
AUC	0.552	0.566	0.799	0.521	0.557	0.510	0.542
Youden index	0.34	0.26	0.58	0.17	0.25	0.24	0.23
Cut-off point	3.04	35.12	1.37	0.74	1.30	1.93	0.89
Sensitivity (%)	100	88.9	88.9	88.9	100	55.6	66.7
Specificity (%)	34.4	37.5	65.6	28.1	25.0	68.8	56.3
Positive predictive value (%)	30.0	28.6	42.1	25.8	27.3	33.3	30.0
Negative predictive value (%)	100.0	92.3	95.5	90.0	100.0	84.6	85.7
Accuracy (%)	48.8	48.8	70.7	41.5	41.5	65.9	58.5
*p*-value	0.5695	0.5044	**0.0006**	0.8369	0.5706	0.9350	0.7210

PAI-1: plasminogen activator inhibitor-1; t-PA: tissue plasminogen activator; u-PA: urokinase plasminogen activator; bold *p*-values denote significant differences.

**Table 5 jcm-11-02398-t005:** Calculated median and ROC cut-off point values of investigated fibrinolytic parameters.

	PAI-1 Activity [U/mL]	PAI-1 Antigen [ng/mL]	t-PA Antigen [ng/mL]	u-PA Activity [U/mL]	u-PA Antigen [ng/mL]	PAI-1/u-PA Complex [ng/mL]	PAI-1/t-PA Complex [ng/mL]
Medians	6.12	48.10	1.41	0.86	0.41	2.65	0.82
ROC cut-off points	3.04	35.12	1.37	1.30	0.74	1.93	0.89

PAI-1: plasminogen activator inhibitor-1; t-PA: tissue plasminogen activator; u-PA: urokinase plasminogen activator.

**Table 6 jcm-11-02398-t006:** Logistic regression analysis of disease-free survival predictors in breast cancer patients.

Variable	Code	Disease-Free Survival	
		OR (95% CI)	*p*-Value
PAI-1 antigen	<48.10 ng/mL ≥48.10 ng/mL	1.25 (0.28–5.53)	0.7685
PAI-1 activity	<6.12 U/mL ≥6.12 U/mL	0.80 (0.18–3.54)	0.7685
u-PA antigen	<0.41 ng/mL ≥0.41 ng/mL	0.71 (0.16–3.12)	0.6461
u-PA activity	<0.86 U/mL ≥0.86 U/mL	1.25 (0.28–5.53)	0.7685
t-PA antigen	<1.41 ng/mL ≥1.41 ng/mL	0.06 (0.01–0.68)	**0.0209**
PAI-1/u-PA complex	<2.65 ng/mL ≥2.65 ng/mL	0.71 (0.16–3.12)	0.6461
PAI-1/t-PA complex	<0.82 ng/mL ≥0.82 ng/mL	2.27 (0.48–10.69)	0.3008

OR: odds ratio; CI: confidence interval; significant differences are denoted by bold *p*-values.

**Table 7 jcm-11-02398-t007:** The multivariate and univariate Cox regression models for disease-free survival.

	Multivariate		Univariate	
Variables	HR (95% CI)	*p*-Values	HR (95% CI)	*p*-Values
PAI-1 antigen				
Low	4.41	0.0768	1.34	0.6652
High	(0.85–22.85)		(0.36–4.99)	
PAI-1 activity				
Low	8.42	0.1497	0.83	0.7784
High	(0.46–152.83)		(0.22–3.08)	
u-PA antigen				
Low	0.73	0.7551	0.74	0.6468
High	(0.10–5.36)		(0.20–2.74)	
u-PA activity				
Low	0.64	0.6849	1.17	0.8192
High	(0.08–5.46)		(0.31–4.34)	
t-PA antigen				
Low	0.15	0.1199	0.10	**0.0323**
High	(0.01–1.64)		(0.01–0.83)	
PAI-1/u-PA complex				
Low	0.12		0.71	
High	(0.01–1.34)	0.0839	(0.19–2.64)	0.6065
PAI-1/t-PA complex				
Low	0.87	0.8812	1.93	0.3527
High	(0.15–5.24)		(0.48–7.72)	

Cox proportional hazards model was used for unadjusted univariate and adjusted multivariate analyses-BMI, age at the time of diagnosis, smoking status, staging, intrinsic type, histological type, nodal involvement and tumour diameter; significant differences are denoted by bold *p*-values.

**Table 8 jcm-11-02398-t008:** Linear regression models for disease-free survival predictors in breast cancer patients.

		Model 1	Model 2	Model 3	Model 4
PAI-1 antigen	Beta *p*-value	0.0914 0.5758	0.1434 0.3886	0.1468 0.3865	0.2002 0.1832
PAI-1 activity	Beta *p*-value	−0.0930 0.5941	−0.0116 0.9556	−0.0094 0.9651	0.3033 0.1133
u-PA antigen	Beta *p*-value	−0.1048 0.6066	−0.0259 0.9793	−0.0204 0.9093	−0.0499 0.7539
u-PA activity	Beta *p*-value	−0.1041 0.5458	−0.1306 0.4624	−0.1386 0.4430	−0.1739 0.2721
t-PA antigen	Beta *p*-value	−0.4197 **0.0071**	−0.3788 **0.0213**	−0.3815 **0.0227**	−0.1996 0.2039
PAI-1/u-PA complex	Beta *p*-value	0.0925 0.5700	0.1224 0.4590	0.1136 0.5064	0.0939 0.5611
PAI-1/t-PA complex	Beta *p*-value	0.0160 0.9228	0.0931 0.5985	0.0836 0.6471	0.1960 0.2081

Model 1 adjusted for age. Model 2 adjusted for age, BMI, parity, menopausal status. Model 3 adjusted for age, BMI, parity, menopausal status and smoking status. Model 4 adjusted for age, BMI, parity, menopausal status, smoking status, tumour stage, tumour diameters, intrinsic type, histological type, nodal involvement. Significant differences are denoted by bold *p*-values.

## Data Availability

The data presented in this study are available in this article.

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
