# Peer review of "Tissue Plasminogen Activator as a Possible Indicator of Breast Cancer Relapse: A Preliminary, Prospective Study"

_jcm, 2022, doi:10.3390/jcm11092398_

Round 1
Reviewer 1 Report
“Tissue plasminogen activator as a possible indicator of breast cancer relapse: A prospective study” by Wrzeszcz et al
In the submitted manuscript, the authors measured PAI-1, u-PA, and t-PA levels in the plasma from a breast cancer patient cohort (n=41) using ELISA kits purchased from a single company. The authors concluded that a low t-PA antigen level or high PAI-1 activity level is a predictor of distant metastasis and poor patient outcomes. The data do not support the conclusion.
The Major problems are:
- No healthy donor controls were included. This significantly limits the authors’ claim on a “diagnostic” value of tPA or any factors for that matter.
- The authors does not explain why both activity vs antigen measurements were performed per a molecule and what the discrepancy signifies. For example, in Table 3, PAI-1 activity was not significantly different between N0 and N1 patient samples (6 vs 6 U/ml), while PAI-1 antigens were significantly different (41 vs 29 ng/ml). Does this mean that there are “inactive” PAI-1 fragments in N0 patient samples? How does this relate to the role of PAI-1 in metastasis? Moreover, the same Table shows that PAI-1/uPA or PAI-1/tPA measurements were not different between N0 and N1 patients. Collectively these make you think that PAI-1 “antigen” measurement may be an artifact of some sort.
- If activity vs antigen measurement comparisons are important, why were tPA “activity” assays not done, only “antigen” assays?
- Table 2 lists the specifications of the ELISA assays used in the study. Were these compiled from the company site or performed by the authors? Why are many of the CV values missing? In these “no data” assay kits (which includes the tPA antigen kit), how did you ensure the quality of the measurements?
- With the intra- and inter-assay coefficient of variations of the ELISA kits are up to 10% (Table 2), it is necessary to present the mean values per patient sample derived from multiple assays to demonstrate reproducibility of the assays.
- Unusual presentation of the measurements in Table 3 as a single mean value followed by what appears to be the range, for example, PAI-1 activity in N0 samples is 6.32 with the min/max values of 1.65/13.12. It is more appropriate to present a mean value with SEM or SD. If the authors want to present min/max values of a wide range, the data should be presented in graphs (e.g. dot plot, box whisker plot, violin plot) with appropriate p-value comparisons.
- The tPA concentration cutoff determined by ROC analysis, 1.37ng/ml, is an arbitrary number since it’s not validated in a separate cohort. At the same time, this ROC analysis does not add any value to the study since the median tPA value 1.41ng/ml also showed significant DFS difference in KM evaluation (Figure 3).
- uPA ROC/KM analysis clearly show a separation in patient survival (Figure 4B). The lack of significance is likely due to the sample size.
- The authors stated that higher PAI-1 “and” lower t-PA correlated with poor patient outcomes. This is not correct since the authors did not present covariate analyses.
- Minor comment: luminal B subtype in the text – lum B tumors are HER2+. There is no HER2-negative lum B.
- Minor comment: Table 7 univariate t-PA HR was 0.1, meaning 90% reduced risk, not 83% (this is for multivariate HR 0.17 which was not significant) as stated in the results and discussion sections. If anything, multivariate HR was not significant, which suggests that t-PA is not predictive of patient outcomes when adjusted for other factors. Though this is likely due to the small cohort size.
In summary, the manuscript conclusions were not supported by the data presented, with the major problems of no healthy donor controls, no assay controls, and no validation cohort. This reviewer recommends reject.
Author Response
Tissue plasminogen activator as a possible indicator of breast cancer relapse: A preliminary, prospective study
Manuscript ID: biology-1591013
We would like to inform you that we have revised our manuscript according to the Reviewers form provided. Below please find attached the list of changes we have introduced.
All modifications have been marked in green.
Reviewer No 1: “Tissue plasminogen activator as a possible indicator of breast cancer relapse: A prospective study” by Wrzeszcz et al
In the submitted manuscript, the authors measured PAI-1, u-PA, and t-PA levels in the plasma from a breast cancer patient cohort (n=41) using ELISA kits purchased from a single company. The authors concluded that a low t-PA antigen level or high PAI-1 activity level is a predictor of distant metastasis and poor patient outcomes. The data do not support the conclusion.
Author: We thank the Reviewer for his/her comments and the suggestions to improve the manuscript.
Reviewer No 1: The Major problems are:
No healthy donor controls were included. This significantly limits the authors’ claim on a “diagnostic” value of tPA or any factors for that matter.
Author: You are right, according to your suggestion we have modified our statement that tPA presents diagnostic value. When designing this study, we did not assume a control group. Since we had access to follow-up data, we wanted to determine the potential prognostic significance of the factors involved in fibrinolysis, applying advanced statistical methods.
Reviewer No 1: The authors does not explain why both activity vs antigen measurements were performed per a molecule and what the discrepancy signifies. For example, in Table 3, PAI-1 activity was not significantly different between N0 and N1 patient samples (6 vs 6 U/ml), while PAI-1 antigens were significantly different (41 vs 29 ng/ml). Does this mean that there are “inactive” PAI-1 fragments in N0 patient samples? How does this relate to the role of PAI-1 in metastasis? Moreover, the same Table shows that PAI-1/uPA or PAI-1/tPA measurements were not different between N0 and N1 patients. Collectively these make you think that PAI-1 “antigen” measurement may be an artifact of some sort.
Author: There are two methods for testing hemostatic parameters, which may be used simultaneously:
Antigen test: This test measures the amount (concentration) of the hemostatic parameter in a blood sample. It is generally reported in milligrams per decilitre (mg/dL) or nanograms per litre (ng/L).
Human uPA total antigen assay is intended for the quantitative determination of total urokinase plasminogen activator in human plasma, serum, urine, cell culture media, or tissue extracts. For research use only. Strip well format. Reagents for up to 96 tests.
Human uPA will bind to the capture antibody coated on the microtiter plate. Free and complexed enzyme will react with the capture antibody on the plate. After appropriate washing steps, polyclonal anti-human uPA primary antibody binds to the captured enzyme. Excess antibody is washed away and bound polyclonal antibody is then reacted with the secondary antibody conjugated to horseradish peroxidase. TMB substrate is used for color development at 450nm. A standard calibration curve is prepared along with the samples to be measured using dilutions of uPA.
However, activity test: This test express the ability to perform hemostatic parameters action (activation or inhibition). Since most of the hemostatic parameters are serine proteases, thus these elements circulate in inactive form.
Functionally active uPA present in samples will form a covalent complex with the biotinylated human PAI-1 which is bound to the avidin on the plate. Inactive or complexed uPA will not bind to the plate and will not be detected. Unbound uPA samples are washed away and an anti-uPA primary antibody is added. Excess primary antibody is washed away and bound antibody is then reacted with the horseradish peroxidase secondary antibody. Following an additional washing step, TMB is then used for color development at 450nm. The amount of color development is directly proportional to the concentration of active uPA in the sample.
Additionally, This human plasminogen activator inhibitor type 1 (PAI-1) total antigen assay is intended for the quantitative determination of total PAI-1 in human plasma in citrate, EDTA or heparin, serum, cell culture media, tissue lysates and other biological fluids. For research use only. Strip well format. Reagents for up to 96 tests.
Human PAI-1 will bind to the monoclonal capture antibody coated on the microtiter plate. Free, latent, and complexed PAI-1 will bind to the plate. After appropriate washing steps, polyclonal anti-human PAI-1 primary antibody binds to the captured protein. Excess primary antibody is washed away and bound antibody is reacted with peroxidase conjugated secondary antibody. Following an additional washing step, TMB substrate is used for color development at 450nm. A standard calibration curve is prepared along with the samples to be measured using dilutions of human PAI-1. Color development is proportional to the concentration of PAI-1 in the samples.
However, This human PAI-1 activity assay is for the quantitative determination of active plasminogen activator inhibitor type 1 (PAI-1) in non-plasma samples including cell culture media or tissue extracts. For research use only. Strip well format. Reagents for up to 96 tests.
Functionally active PAI-1 present in samples reacts with urokinase coated and dried on a microtiter plate. Latent or complexed PAI-1 will not bind to the plate and will not be detected. After appropriate washing steps, monoclonal anti-human PAI-1 primary antibody binds to the captured enzyme. Excess antibody is washed away and bound monoclonal antibody is then reacted with the secondary antibody conjugated to horseradish peroxidase. TMB substrate is used for color development at 450nm. Color development is directly proportional to the concentration of active PAI-1 in the samples. A standard calibration curve is prepared along with the samples to be measured using dilutions of human PAI-1.
Reviewer No 1: If activity vs antigen measurement comparisons are important, why were tPA “activity” assays not done, only “antigen” assays?
Author: During study designing, we assumed the determination of the activity and concentration of t-PA. However, we have met methodological issues, since blood for t-PA activity should be acidified during blood collection, but hospital conditions did not allow for such additional activity.
Reviewer No 1: Table 2 lists the specifications of the ELISA assays used in the study. Were these compiled from the company site or performed by the authors? Why are many of the CV values missing? In these “no data” assay kits (which includes the tPA antigen kit), how did you ensure the quality of the measurements?
Author: According to your suggestion you have typed an email to the Molecular Innovations Company, Inc., Novi, MI, USA. Thanks to this action we have received following answers:
Our query: In our research, we used ELISA kits manufactured by your company. I want to inquire about the CV values (intra- and interassay) for the following reagents:
- Human u-PA Total Antigen ELISA Kit
- Human t-PA Total Antigen ELISA Kit
- Human PAI-1/u-PA 1 Complex Antigen ELISA Kit
- Human PAI-1/t-PA Complex Antigen ELISA Kit
Unfortunately, this information is not included in the leaflet, and we need it to publish the manuscript. I will be grateful if you can also follow the literature in which these tests have already been used. In the previous version of the website, such data was given, but now, after the merger of your company with Innovative Research, I cannot find such data. Thank you very much; I hope your e-mail addresses still work.
Company response: Dear Dr. Slomka,
Thank you for reaching out to us. We don’t have all the performance characteristics completed, but I am including the information that we have completed for each specific ELISA kit in your request:
- Human u-PA Total Antigen ELISA Kit Intra-assay Precision: Three samples of known concentration were tested twenty times on one plate to assess intra-assay precision.
Inter-assay Precision: These studies are currently in progress. For each of the following 3 kits,
- Human t-PA Total Antigen ELISA Kit
- Human PAI-1/u-PA 1 Complex Antigen ELISA Kit
- Human PAI-1/t-PA Complex Antigen ELISA Kit
The Intra-assay Precision and Inter-assay Precision: These studies are currently in progress.
We don’t have any more information at this time, but we can make these kits more a priority and will let you when they are complete.
Tabitha M. Doci
ELISA Manager
P: 248-896-0145
F: 248-896-0149
E: tdoci@innov-research.com
Reviewer No 1: With the intra- and inter-assay coefficient of variations of the ELISA kits are up to 10% (Table 2), it is necessary to present the mean values per patient sample derived from multiple assays to demonstrate reproducibility of the assays.
Author: According to your suggestion you have typed an email to the Molecular Innovations Company, Inc., Novi, MI, USA. Thanks to this action we have received following answers:
Our query: Dear Tabitha,
Thank you very much for your quick and precise answer.
Based on your research, can we expect that inter-assay %CV will be less than 15%, while intra-assay %CV will be less than 10%.
Warmest wishes,
Artur Słomka
Company response: Dear Artur,
For each of the Performance Characteristics, we follow a standard procedure with a
specific set of acceptable values, for example:
Intra-assay Precision: Three samples of known concentration were tested twenty times on one plate to assess intra-assay precision. Acceptable values are %CV <10%
Inter-assay Precision: Three samples of known concentration were tested in ten
independent assays to assess inter-assay precision. Acceptable values are %CV <15%
When we do complete these procedures for these specific kits, we feel confident that they will also pass the criteria we established.
I hope this helps with your studies.
Thank you,
Tabitha M. Doci
ELISA Manager
Thus, we have provided intra-assay and inter-assay coefficients of variation in Table 3.
Reviewer No 1: Unusual presentation of the measurements in Table 3 as a single mean value followed by what appears to be the range, for example, PAI-1 activity in N0 samples is 6.32 with the min/max values of 1.65/13.12. It is more appropriate to present a mean value with SEM or SD. If the authors want to present min/max values of a wide range, the data should be presented in graphs (e.g. dot plot, box whisker plot, violin plot) with appropriate p-value comparisons.
Author: The normality of the variables was tested by the Shapiro–Wilk test. Based on the Shapiro-Wilk we demonstrated that data tested in our study are not normally-distributed. Therefore, to analyze the difference in the expression of two independent, not normally-distributed variables we used U Manna-Whitney test, and we presented data as medians (Me) and interquartile ranges (Q1 and Q3).
Reviewer No 1: The tPA concentration cutoff determined by ROC analysis, 1.37ng/ml, is an arbitrary number since it’s not validated in a separate cohort. At the same time, this ROC analysis does not add any value to the study since the median tPA value 1.41ng/ml also showed significant DFS difference in KM evaluation (Figure 3).
Author: In respect to this issue we would like to inform you that all ROC cut-off points include area under the curve (AUC), Youden index, Sensitivity (%), Specificity (%), also Accuracy (%). You are right that these cut-off points should be confirmed in larger, adequately-powered study.
ROC data |
Stimulant |
Stimulant |
Destimulant |
Destimulant |
Destimulant |
Destimulant |
Stimulant |
PAI-1 activity |
PAI-1 antigen |
t-PA antigen |
u-PA antigen |
u-PA activity |
PAI-1/ |
PAI-1/ |
|
AUC |
0.552 |
0.566 |
0.799 |
0.521 |
0.557 |
0.510 |
0.542 |
Youden index |
0.34 |
0.26 |
0.58 |
0.17 |
0.25 |
0.24 |
0.23 |
Cut-off point |
3.04 |
35.12 |
1.37 |
0.74 |
1.30 |
1.93 |
0.89 |
Sensitivity (%) |
100 |
88.9 |
88.9 |
88.9 |
100 |
55.6 |
66.7 |
Specificity (%) |
34.4 |
37.5 |
65.6 |
28.1 |
25.0 |
68.8 |
56.3 |
Positive predictive value (%) |
30.0 |
28.6 |
42.1 |
25.8 |
27.3 |
33.3 |
30.0 |
Negative predictive value (%) |
100.0 |
92.3 |
95.5 |
90.0 |
100.0 |
84.6 |
85.7 |
Accuracy (%) |
48.8 |
48.8 |
70.7 |
41.5 |
41.5 |
65.9 |
58.5 |
P-value |
0.5695 |
0.5044 |
0.0006 |
0.8369 |
0.5706 |
0.9350 |
0.7210 |
Nevertheless only t-PA cut-off points and PAI-1/t-PA complex were similar in respect to median and ROC curve. Thus we applied both cut-off points in Kaplan-Meier analysis.
|
PAI-1 activity [U/mL] |
PAI-1 antigen [ng/mL] |
t-PA antigen [ng/mL] |
u-PA activity [U/mL] |
u-PA antigen [ng/mL] |
PAI-1/ complex [ng/mL] |
PAI-1/ |
|
Medians |
6.12 |
48.10 |
1.41 |
0.86 |
0.41 |
2.65 |
0.82 |
|
ROC cut-off points |
3.04 |
35.12 |
1.37 |
1.30 |
0.74 |
1.93 |
0.89 |
|
Reviewer No 1: uPA ROC/KM analysis clearly show a separation in patient survival (Figure 4B). The lack of significance is likely due to the sample size.
Author: According to your suggestion we have added following information in this regard:
The Kaplan-Meier analysis revealed an essential growing tendency towards a higher risk of disease relapse in patients with lower activity of u-PA than 1.3 U/mL (p=0.0969). This observation demonstrates a similar trend of u-PA activity as the t-PA antigen. Most likely, the lack of statistical difference may be due to the small size of the study group. The study also indicates that patients with early-stage of breast cancer demonstrate more frequently lower u-PA activity (33 cases (80%) with lower vs 8 (20%) with higher u-PA activity than 1.30 U/mL).
Reviewer No 1: The authors stated that higher PAI-1 “and” lower t-PA correlated with poor patient outcomes. This is not correct since the authors did not present covariate analyses.
Author: You are absolutely right we have modified this sentence.
Reviewer No 1: Minor comment: luminal B subtype in the text – lum B tumors are HER2+. There is no HER2-negative lum B.
Author: Perhaps the discrepancies in the nomenclature are due to the fact that patients were qualified for the study between 2015 and 2017 and recommendations by St. Gallen form 2011 were applied:
In 2015-2017 in Oncology Centre in Bydgoszcz, Poland, biological (molecular) subtypes of Invasive Breast Cancer as recommended by St. Gallen 2011:
- luminal A - ER, PrR +; HER2 -; Ki-67 <20%
- luminal B (HER2 negative) - ER, PrR +; HER2 -; Ki-67> 20% ( or PrR <20%
- luminal B (HER2 positive) - ER, PrR, HER2 +; Ki-67 - each
- HER2-positive (non-luminal) - ER, PrR, -; HER2 +; Ki-67 - each
- three-negative (basal, cable) - ER, PrR, HER2 -; Ki-67 - each
- special molecular (biological) types of breast cancer [crush, tubular, mucous, apocrine, medullary, glandular cystic, metaplastic carcinoma]: - hormone-dependent - ER, PrR +; HER2 -; Ki-67 - each, - hormone-independent - ER, PrR -; HER2 -; Ki-67 - each.
Based on it in our study population there were 26 patients with luminal A IBrC, 7 with luminal B (HER2-) but 5 with luminal B (HER2+) and 3 with triple-negative.
Reviewer No 1: Minor comment: Table 7 univariate t-PA HR was 0.1, meaning 90% reduced risk, not 83% (this is for multivariate HR 0.17 which was not significant) as stated in the results and discussion sections. If anything, multivariate HR was not significant, which suggests that t-PA is not predictive of patient outcomes when adjusted for other factors. Though this is likely due to the small cohort size.
Author: You are absolutely right, according to you suggestion we have corrected this mistake.
Reviewer No 1: In summary, the manuscript conclusions were not supported by the data presented, with the major problems of no healthy donor controls, no assay controls, and no validation cohort. This reviewer recommends reject.
Author: We are fully aware that the study population is quite small. Thus we have modified the title of the manuscript into 'Tissue plasminogen activator as a possible indicator of breast cancer relapse: A preliminary, prospective study'.
Also we have provided the limitation of the study:
There are some limitations of our study. First, we included a small number of patients, which could constitute a bias; data should be confirmed in a more extensive and prospective series of patients. Although we collected and adjusted data for probable confounders in our study, there could be unmeasured and undefined factors with possible residual effects. We evaluated patients in single-centre designed research. Further prospective studies with more significant numbers and longer follow-ups are required to investigate the long-term outcome. Sufficient evidence has not yet been accumulated to determine cut-off values of the t-PA, u-PA, PAI-1 or PAI-1/t-PA and PAI-1/uPA complexes for early-stage IBrC; thus, further evaluation should be performed.
Reviewer 2 Report
This is a nicely designed prospective study of tissue plasminogen activator as a possible indicator of breast cancer relapse. The findings demonstrate that t-PA antigen levels may serve as prognostic markers for early breast cancer. Below are a few comments for the authors consideration.
- There are many errors of English and grammar in the manuscript. The help of an English editor may be solicited for revision.
- What was the race/ethnicity of the study participants? This should be stated in the paper and included in table 1. It is a documented fact that race plays a key role in breast cancer histochemistry and prognosis.
- Why was smoking status not included as part of the covariates adjusted in the Cox regression model. Cigarette smoking is associated with increased fibrinogen levels, unaltered fibrinolysis, and overall survival.
- See highlighted text of some errors in English in the uploaded file for author convenience during revision.

Author Response
Tissue plasminogen activator as a possible indicator of breast cancer relapse: A preliminary, prospective study
Manuscript ID: jcm-1635435
All modifications have been marked in yellow.
Reviewer No 2: This is a nicely designed prospective study of tissue plasminogen activator as a possible indicator of breast cancer relapse. The findings demonstrate that t-PA antigen levels may serve as prognostic markers for early breast cancer. Below are a few comments for the authors consideration.
Author: We thank the Reviewer for his/her positive comments. We wish you all the best. We appreciate your kindness.
Reviewer No 2: There are many errors of English and grammar in the manuscript. The help of an English editor may be solicited for revision.
Author: The style and language have been improved by a professional English translator to enhance reader-friendliness.
Reviewer No 2: What was the race/ethnicity of the study participants? This should be stated in the paper and included in table 1. It is a documented fact that race plays a key role in breast cancer histochemistry and prognosis.
Author: According to your suggestion we have added information related to race/ethnicity of the study participants. All subjects were of Polish descent. We have also provided in Table 1 information: Total (n = 41; 100% of European ancestry).
Reviewer No 2: Why was smoking status not included as part of the covariates adjusted in the Cox regression model. Cigarette smoking is associated with increased fibrinogen levels, unaltered fibrinolysis, and overall survival.
Author: According to your suggestion we have recalculated data in Table 7 and 8 in order to include smoking status.
In this study we also used a univariate and multivariate Cox regression to analyse the prognostic factors of disease-free survival, which takes into account the function of time. The multivariate Cox regression model was adjusted for prognostic factors including BMI, age at the time of diagnosis, smoking status, staging system, intrinsic type, histological type, nodal involvement and tumour diameter (Table 7). Thus, in a multivariate analysis, by including fibrinolytic parameters in a model with all the traditional predictive factors, we found no significant association of fibrinolytic parameters levels with the risk of breast cancer relapse (p>0.05). However, the univariate Cox regression model showed that t-PA antigen was significantly associated with a prolonged DFS (HR = 0.10, 95% CI = 0.01–0.83, p= 0.0323). According to these results, patients with a t-PA antigen higher than 1.41 ng/mL appear to have a 90% decreased risk of breast cancer recurrence.
Table 7. The multivariate and univariate Cox regression models for disease-free survival.
|
Multivariate |
Univariate |
||
Variables |
HR (95% CI) |
P-values |
HR (95% CI) |
P-values |
PAI-1 antigen Low High |
4.41 (0.85-22.85) |
0.0768 |
1.34 (0.36-4.99) |
0.6652 |
PAI-1 activity Low High |
8.42 (0.46-152.83) |
0.1497 |
0.83 (0.22-3.08) |
0.7784 |
u-PA antigen Low High |
0.73 (0.10-5.36) |
0.7551 |
0.74 (0.20-2.74) |
0.6468 |
u-PA activity Low High |
0.64 (0.08-5.46) |
0.6849 |
1.17 (0.31-4.34) |
0.8192 |
t-PA antigen Low High |
0.15 (0.01-1.64) |
0.1199 |
0.10 (0.01-0.83) |
0.0323 |
PAI-1/u-PA complex Low High |
0.12 (0.01-1.34) |
0.0839 |
0.71 (0.19-2.64) |
0.6065 |
PAI-1/t-PA complex Low High |
0.87 (0.15-5.24) |
0.8812 |
1.93 (0.48-7.72) |
0.3527 |
Cox proportional hazards model was used for unadjusted univariate and adjusted multivariate analyses- BMI, age at the time of diagnosis, smoking status, staging, intrinsic type, histological type, nodal involvement and tumour diameter; significant differences are denoted by bold p-values.
The associations of fibrinolytic parameters level and the disease-free survival by multiple linear regression analyses are shown in Table 8. The breast cancer recurrence was negatively associated only with t-PA antigen concentration using multivariate linear regression analyses after adjusting for age, BMI, parity, menopausal status and smoking status. In model 1 adjusted for age, the results showed that a lower t-PA antigen level was correlated with a higher risk of breast cancer relapse (standardized Beta = -0.4197; p=0.0071). Similarly, in model 2 and 3, the results showed that a lower t-PA antigen level was associated with a higher risk of relapse occurrence after adjusting for age, BMI, parity, menopausal status and smoking status (standardized Beta = -0.3788; p=0.0213 for model 2 and standardized Beta = -0.3815; p=0.0227 for model 3, respectively). After adjusting for age, BMI, parity, menopausal status, smoking status, tumour stage, tumour diameters, intrinsic type, histological type, and nodal involvement (model 4) we found no significant trend of breast cancer relapse and t-PA antigen concentration (p>0.05).
Table 8. Linear regression models for disease-free survival predictors in breast cancer patients.
|
Model 1 |
Model 2 |
Model 3 |
Model 4 |
|
PAI-1 antigen
|
Beta P-value |
0.0914 0.5758 |
0.1434 0.3886 |
0.1468 0.3865 |
0.2002 0.1832 |
PAI-1 activity
|
Beta P-value |
-0.0930 0.5941 |
-0.0116 0.9556 |
-0.0094 0.9651 |
0.3033 0.1133 |
u-PA antigen
|
Beta P-value |
-0.1048 0.6066 |
-0.0259 0.9793 |
-0.0204 0.9093 |
-0.0499 0.7539 |
u-PA activity
|
Beta P-value |
-0.1041 0.5458 |
-0.1306 0.4624 |
-0.1386 0.4430 |
-0.1739 0.2721 |
t-PA antigen
|
Beta P-value |
-0.4197 0.0071 |
-0.3788 0.0213 |
-0.3815 0.0227 |
-0.1996 0.2039 |
PAI-1/u-PA complex |
Beta P-value |
0.0925 0.5700 |
0.1224 0.4590 |
0.1136 0.5064 |
0.0939 0.5611 |
PAI-1/t-PA complex |
Beta P-value |
0.0160 0.9228 |
0.0931 0.5985 |
0.0836 0.6471 |
0.1960 0.2081 |
Model 1 adjusted for age. Model 2 adjusted for age, BMI, parity, menopausal status. Model 3 adjusted for age, BMI, parity, menopausal status and smoking status. Model 4 adjusted for age, BMI, parity, menopausal status, smoking status, tumour stage, tumour diameters, intrinsic type, histological type, nodal involvement. Significant differences are denoted by bold p-values.
Reviewer No 2: See highlighted text of some errors in English in the uploaded file for author convenience during revision.
Author: Thank you for drawing attention of these issues. We would like to inform you that the style and language have been improved by a professional English translator to enhance reader-friendliness.
Round 2
Reviewer 2 Report
The authors have adequately responded to my comments from the initial review. Best of luck!